# Trends from the Last Decade with Nontuberculous Mycobacteria Lung Disease (NTM-LD): Clinicians’ Perspectives in Regional Center of Pulmonology in Bydgoszcz, Poland

**DOI:** 10.3390/pathogens12080988

**Published:** 2023-07-28

**Authors:** Grzegorz Przybylski, Jakub Bukowski, Weronika Kowalska, Marta Pilaczyńska-Cemel, Dorota Krawiecka

**Affiliations:** 1Department of Respiratory Medicine and Lung Diseases, Collegium Medicum in Bydgoszcz, Nicolaus Copernicus in Torun, 87-100 Toruń, Poland; kowalskaweronika555@gmail.com (W.K.); mpilaczynska@wp.pl (M.P.-C.); 2Regional Center of Pulmonology in Bydgoszcz, 85-326 Bydgoszcz, Poland; j_bukowski@o2.pl (J.B.); dorota.krawiecka@gmail.com (D.K.)

**Keywords:** nontuberculous mycobacteria (NTM), mycobacteria other than tuberculosis (MOTT), nontuberculous mycobacteria lung disease (NTM-LD), epidemiology, microbiological diagnostics, radiological findings, symptoms

## Abstract

Background: Nontuberculous mycobacteria (NTM) are the cause of chronic lung disease called NTM lung disease (NTM-LD). There are about 180 known species of NTM. Nowadays the number of NTM-LD is increasing. Objective: To evaluate the clinical significance of NTM isolated from specimens and assess the frequency and clinical relevance of isolation of NTM in the Regional Center of Pulmonology in Bydgoszcz, hospital of Northern Poland. Design: Clinical, radiological, and microbiological data were collected from all patients from whom NTM was isolated between 2013 and 2022. Data were reviewed retrospectively. Diagnostic criteria for NTM-LD published by the American Thoracic Society (ATS) were used to determine clinical relevance. Material and methods: The study comprised 81,985 clinical specimens submitted for mycobacterial culture in the Department of Microbiology at the Regional Center of Pulmonology in Bydgoszcz between 2013 and 2022. Clinical specimens were processed according to the standard procedure in mycobacteria laboratories in Poland. NTM strains were identified using analysis of mycolic acids by chromatography as well as GenoType NTM-DR, GenoType Mycobacterium AS, and GenoType Mycobacterium CM. Results: There were 395 patients with NTM strains between 2013 and 2022. Out of them, 149 cases met the diagnostic criteria of NTM-LD and were classified as definite cases. *M. kansasii* (n = 77) was the most common species in the group (51.68%), followed by *M. avium* complex (n = 46). Patients with NTM-LD were 22–88 years old (median age was 60 years). There were 81 men and 68 women. The most common symptoms were cough, hemoptysis, and fever. Radiological X-ray images were dominated by infiltrative lesions in the upper and middle lobe of the right lung with cavities; the changes were in the upper lobe of the left lung and on both sides of the chest. They were smokers in 61%. The most common concomitant diseases were chronic obstructive pulmonary disease (COPD), diabetes mellitus, pulmonary carcinoma, and human immunodeficiency virus (HIV) infection, and other immunodeficiencies. The most common treatment was isoniazid, ethambutol, rifampicin, and ofloxacin for 18 months with a minimum of 12 months of culture negativity. Conclusions: NTM-LD infections are present with other pulmonary illnesses and extrapulmonary diseases and may be connected to primary immunologic deficiencies. These diseases concern patients of all ages and have various clinical manifestations. *M. kansasii* and MAC are the most prevalent NTM isolates among respiratory samples in Northern Poland. In addition, an increase in MAC and a decrease in *M. kansasii* both in cultivation and the cause of NTM-LD were reported.

## 1. Introduction

Nontuberculous mycobacteria are ubiquitous environmental organisms in soil, water, and medical devices such as bronchoscopes [1]. Over 180 NTM species have been identified [2]. It is a heterogeneous group of opportunistic pathogens causing human disease predominantly in immunocompromised individuals and patients with underlying diseases such as bronchiectasis and cystic fibrosis [3,4]. Clinical diseases caused by NTM can be very diverse and present with pulmonary lesions and involve extrapulmonary sites such as lymph nodes, skin and soft tissue, bones, and joints [5]. NTM disease can also mimic infection caused by the *Mycobacterium tuberculosis* complex (MTBC) [2].

Several studies have reported an increase in both NTM diseases and NTM colonization in humans during the last decades [6,7,8,9,10,11]. The increase in NTM diseases worldwide has been attributed to different factors such as improved culturing techniques and greater disease awareness, but it has also been attributed to a true increase in disease incidence as a result of increased life expectancy and an increasing proportion of the population experiencing immunosuppression due to medication or suffering from comorbidities such as, e.g., diabetes mellitus and COPD [12,13,14].

The prevalence of pulmonary NTM diseases is approximately 1–15 per 100,000 person-years in the United States [8]. The prevalence of NTM disease per 100,000 people in South Korea increased from 9.4 in 2009 to 36.1 in 2016 [15]. The reported incidence of NTM lung diseases in South Korea has risen from 1.2 to 4.8 in 2016 [16]. Prevalence of NTM disease in Germany documented prevalence rates of this non-notifiable disease increased from 2.3 to 3.3 cases/100,000 population from 2009 to 2014 [9]. The prevalence of NTM diseases per 100,000 people in Poland was 0.61 in 2019 [17].

NTM are most frequently isolated from pulmonary sites and less commonly isolated from extrapulmonary sites such as cerebrospinal fluid, joint fluid, soft tissue, muscle, and bone [18]. The isolation rates of pulmonary disease vary by geographic area, race, and underlying conditions [19].

Among adult patients with NTM diseases, COPD is the most common underlying condition [6,20,21]. Pulmonary disease caused by NTM is commonly encountered in middle-aged men with chronic lung disease and upper lobe cavitations and nodules. NTM-LD is also seen in elderly female patients with no preexisting lung disease. NTM are frequently isolated from respiratory samples. In addition to microbiological results, the diagnosis of NTM-LD should also be based on clinical and radiographic criteria. In 2007, the American Thoracic Society (ATS) and the Infectious Disease Society of America (IDSA) jointly established a set of diagnostic criteria for NTM-LD. The diagnosis is based on characteristic radiographic findings and positive culture results [14]. In 2020, the ATS/European Respiratory Society (ERS)/European Society of Clinical Microbiology and Infectious Diseases (ESCMID)/IDSA guidelines were published, and the updated guidelines recommend the use of the same diagnostic criteria as before [22].

We performed a retrospective case study to assess the prevalence and clinical relevance of NTM isolated in the Regional Center of Pulmonology in Bydgoszcz, Poland. We aimed to evaluate the epidemiology and clinical significance of the pathogens isolated according to ATS/ IDSA criteria.

## 2. Materials and Methods

### 2.1. Study Subjects

We performed a retrospective analysis of microbiological testing results for mycobacteria in Department of Microbiology at Regional Center of Pulmonology in Bydgoszcz from January 2013 to December 2022. The cultured mycobacteria were identified to the species. Further clinical and epidemiological case studies focused on all patients with at least one NTM isolate. In case of multiple isolates of the same species from a single patient, only the first isolate was considered. After identification of NTM, the records of all patients were reviewed. The following data were collected from the patient’s charts: age, clinical presentation, underlying diseases, and treatment.

In 2007, the American Thoracic Society (ATS) and Infectious Disease Society of America (IDSA) jointly established a set of diagnostic criteria for NTM lung disease [14]. The diagnosis is based on characteristic radiographic findings and positive culture results from at least two separate positive cultures of sputum samples, or at least one positive culture from bronchial lavage or lung biopsy [14,22].

### 2.2. Laboratory Procedures

The methods used in Department of Microbiology were in accordance with the methodology in force in mycobacteria laboratories in Poland [23]. During 10-year study period, there were 81,985 specimens submitted for mycobacterial culture. The annual numbers of assays are shown in Figure 1. It shows a similar number of assays until 2019 and a significant decrease from 2020 to 2022 during the COVID-19 pandemic.

Many different types of clinical specimens were collected for analyses, the majority from the respiratory tract (sputum and BAL fluids). Urine, gastric aspirates, tissue fragments, biopsy specimens, normally sterile body fluids, and blood were other submitted specimens. The majority of specimens were non-sterile and needed to be decontaminated and concentrated by centrifugation. The processed sediments were inoculated to growth media specially used for culture of *Mycobacterium* species, Lӧwenstein–Jensen solid medium observed weekly for 10 weeks and Middlebrook 7H9 liquid medium incubated for 6 weeks maximum in BACTEC™ MGIT 960 System (Becton Dickinson, Sparks, Maryland USA). Simultaneously, smear microscopy was performed. Each smear from a clinical specimen was stained using the Ziehl–Neelsen method and thoroughly examined for the presence of acid-fast bacilli (AFB). BACTEC™ Myco/F Lytic culture medium incubated for 6 weeks maximum in BD BACTEC™ FX Blood Culture System (Becton Dickinson, Sparks, Maryland USA) was used for the detection of mycobacteria from blood. Cultured mycobacteria were assigned to MTBC or to NTM using BD MGIT TBc Identification Test (Becton Dickinson, Sparks, Maryland USA). Next, NTM isolates were identified by analysis of mycolic acids by chromatography as well as by genetic investigations using molecular methods: GenoType NTM-DR, GenoType Mycobacterium AS, and GenoType Mycobacterium CM assays (Hain Lifescience, Nehren, Germany).

## 3. Results

### 3.1. Microbiological Examination

Over the 10 years, there were 3104 patients with mycobacteria strains. These included 2709 patients with MTBC strains (87.3% of all patients) and 395 with NTM strains (12.7%). A decrease in number of patients with cultured MTBC was observed from 386 in 2013 to 241 in 2022. The number of patients with cultured NTM remained similar with a peak in 2018 (n = 57). The proportion of patients with MTBC and NTM strains was compared (annual percentage changes). A slight upward trend of patients with cultured NTM among all patients with mycobacteria was noted. The numbers of patients with MTBC and NTM strains are shown year by year in Figure 2.

All of the patients with cultured MTBC were considered tuberculosis cases, while all of the patients with cultured NTM required further analysis to be considered NTM-LD.

All 395 NTM strains were cultured from sputum (n = 238), BAL fluids (n = 136), tissue fragments (n = 16), blood (n = 4), or pleural effusion (n =1). Blood samples were collected from patients infected with HIV.

*M. kansasii* was the most commonly detected NTM (n = 134, which is 34% of all NTM strains), followed by *M. avium* (n = 117), *M. gordonae* (n = 39), rapidly growing mycobacteria RGM (n = 38), *M. intracellulare* (n = 22), *M. xenopi* (n = 21), *M. chimaera* (n = 9) and *M. malmoense* (n = 7). RGM included *M. fortuitum* (n = 11), *M. chelonae* (n = 3), *M. peregrinum* (n = 3), *M. abscessus* (n = 2), *M. mageritense* (n = 1) and *M. neoaurum* (n = 1). A total of 17 RGM strains were not identified at the species level due to clinical decisions. All cultured NTM strains are shown in Table 1.

There were 148 MAC strains, including *M. avium*, *M. intracellulare,* and *M. chimaera*. Numbers of MAC and *M. kansasii* during the 10-year analysis period were compared. *M. kansasii* was the most commonly cultured NTM until 2017. A decrease in its cultivation was observed from 2019 to 2022. *M. avium* complex was the most commonly cultured NTM since 2018. Numbers of MAC and *M. kansasii* strains are shown year by year in Figure 3.

All of the 395 patients with NTM strains were evaluated. Clinical and radiographic criteria of NTM-LD were met in 149 cases, and every case was considered “definite”. The rest of the cases (n = 246) did not meet the criteria of NTM-LD according to ATS. The numbers of patients with NTM-LD among all patients with cultured NTM are shown year by year in Figure 4.

There were 77 patients with *M. kansasii* strains (51.68% of all NTM-LD cases), followed by patients with MAC (n = 46), *M. xenopi* (n = 13), *M. malmoense* (n = 5), *M. gordonae* (n = 3), *M. szulgai* (n = 2), *M. abscessus* (n = 1), *M. fortuitum* (n = 1) and *M. interjectum* (n = 1). The numbers of NTM species among patients with NTM-LD are shown year by year in Table 2.

### 3.2. Clinical and Radiological Significance

The patient age range was 22–88 years (median 60 years); 68 (45.6%) were female and 81 (54.4%) were male. Data were analyzed in the system of selected underlying conditions. Forty-one (27.5%) of the patients had underlying pulmonary disease, e.g., COPD, asthma, and bronchiectasis. Nicotinism was the most common chronic condition. Over 60% of patients were active smokers. Over 10% of subjects had tuberculosis at some point in their life. There were similar components of patients suffering from cancer and diabetes (less than 10% of subjects). In the group of ten patients with underlying cancer, six of them had lung cancer, three of them had hematological malignancies, and they were included in the group of patients with immunodeficiency. There was one patient with colorectal cancer. There were seven immunocompromised patients. It includes patients with leukemia, HIV infection, and patients during biological treatment. Only two of four patients with HIV infection were included. One of the two not included patients died before starting the treatment. The second was lost to follow-up. These data are distorted, because there is a different health center for patients HIV positive in Bydgoszcz; this is why there are so few of them in our research. Precise data about underlying conditions are shown in Table 3. Despite that clinical pictures of mycobacteriosis were different and there were various symptoms and medical backgrounds of the patients, the majority of them were treated with the same combination of medication: rifampicin, isoniazid, and ethambutol. This grouping usually gave a good treatment result. The radiological changes coexisted with clinical improvement and usually occurred after 3–4 months. In the majority, there was good tolerance to the treatment, and there were only two cases of serious side effects, like central nervous system disorders or blood and lymphatic system disorders. The shortest treatment duration was 16 months and the longest was 22 months (median 18 months).

In this study, the presence of clinical changes was considered if the patient reported them in a medical interview, or if they were observed during the examination or hospitalization. During the research period, the most common symptom was a cough, and this has not changed during the last decade. Cough is not as common as it was in 2013—in that year, 87.5% of patients had suffered from this symptom, but only 66.7% in the last year of the research. It is worth mentioning that in the group of 118 patients with a cough, 29 had underlying lung disease, e.g., bronchiectasis, COPD, and asthma. This is why it is difficult to determine whether the cough in this group resulted from mycobacteriosis or from a concomitant disease. In 45% of cases with a cough, this symptom, although less severe, lasted longer than a year, despite the right treatment. Fever was present in about 24% (36 of 149 patients) of all cases of mycobacteriosis. Only two patients suffering from mycobacteriosis with a fever had comorbidities like cancer or immune disorder. In the middle of the research, there was a decline in the number of patients with fever, but in the last three years of this study, the proportion of patients with this syndrome returned to a similar level as at the beginning of the research. Significant change is observed in the proportion of patients with hemoptysis: from only 6.25% in 2013 to 33.3% ten years later. However, this is the least common manifestation of mycobacteriosis—less than 17% of observed cases were presented with hemoptysis. There is not much difference between the presence of two or more symptoms in the group of people younger than 55 years old (16 out of 49) and older than 55 years old (31 out of 100). There is also a similar ratio of polysymptomatic mycobacteriosis between men (26 out of 81) and women (21 out of 68).

When diagnosing NTM-LD, each patient had a chest X-ray and if the changes were difficult to define or very advanced, then the patient had a chest CT. Over the years there are more often cases of mycobacteriosis with caves and involvement of both lungs. In comparison, there were about 20% more patients with caves in their lungs in 2022 than in 2013. The average percent of involvement of both lungs from the last three years of the research is 68.2% compared to 58.8% from the first three years of the survey.

For the eight years of the research involvement of both lungs was more common than the presence of the caves. This proportion changed in 2021, and the next year caves were a more common radiological image of mycobacteriosis (Figure 5). It should be observed whether this proportion will be maintained or become more pronounced in the coming years, or whether the ratio will approach the situation from previous years.

There is a noticeable difference between MAC and *M. kansasii* infections in the radiological picture. In MAC infection in only 22% of cases, there were caves, and almost twice as many (39%) caves were present in *M. kansasii* infections. Involvement of both lungs was similarly often, irrelevant to the pathogen (67% in MAC infections and 69% in *M. kansasii* infections).

The presence of single, multiple nodular lesions or fibrous cavernous malformations was observed in almost 100% of cases (147 of 149 patients). A great advantage of involvement of upper lobes than any other combination was observed. Lower lobes are involved the least number of times and that is only 9.2% of cases during the decade.

### 3.3. Trends

During the 10-year analysis period, the annual number of specimens submitted for mycobacterial culture has been decreasing, especially between 2020 and 2022. A slightly increasing trend of patients with NTM isolates among all patients with mycobacteria was reported. *M. kansasii* and MAC were the most commonly detected NTM. There is a decreasing trend of patients with *M. kansasii* strains and an increasing trend of patients with MAC strains.

Over the years there are more often cases of NTM-LD with caves and involvement of both lungs. In comparison, there were about 20% more patients with caves in their lungs in 2022 than in 2013. The average percent of involvement of both lungs from the last three years of the research is 68.2% compared to 58.8% from the first three years of the survey. The most common symptom is a cough, and this has not changed during the last decade.

## 4. Discussion

Increased awareness of NTM-LD has led to efforts to more fully describe the burden, epidemiological and clinical features, and cost of this condition in different populations globally. In Poland, NTM-LD cases are mandatorily reportable, however, there is no species-specific registry [17]. The epidemiology of NTM infections is poorly understood. Both between and within countries, the geographic diversity is high and many studies on environmental factors (e.g., climate, agriculture, socioeconomic) in NTM transmission have investigated the reasons for the pronounced geospatial variation [24,25,26,27]. In many regions worldwide, the incidence of NTM has been reported to increase for both infection and disease [27,28,29,30]. An increase has also been observed among susceptible high-risk populations, such as patients with cystic fibrosis, non-cystic fibrosis bronchiectasis, and chronic obstructive pulmonary disease, among others [31,32,33]. In selected countries in Europe, the availability of a centralized public health laboratory has served as a tremendous resource for describing the epidemiology of NTM-LD [34,35,36] In the UK, five mycobacteriology reference laboratories serve England, Wales, and Northern Ireland, all of which report to Public Health England. Although detailed clinical information is not available, these data are useful in describing burdens and trends. Based on numerous clinical specimens analyzed for mycobacteria during a quarter century in Poland, we cannot confirm the numerous reports on the increasing incidence of NTM disease. We need uniform reporting of NTM disease. Our analysis for 2013–2022 identified a doubling of the incidence of pulmonary MAC (Figure 3), continuing the trend reported previously [34,35]. In Scotland, although no temporal trend was observed, an increase in *M. avium* was noted during the period 2000–2010 [36]. Denmark has a national registry maintained at the Statens Serum Institute, holding data from all patients with NTM-positive specimens, which has been used to estimate incidence, prevalence, and mortality. A key aspect of the study was the linkage of microbiology information to the Danish National Registry of Patients, which allowed the estimation of infection and disease rates by comorbidity status, and of 3- and 5-year cumulative survival by infecting species [37].

NTM pulmonary isolates are encountered with increasing frequency worldwide. The growing availability of molecular-based detection has contributed to this trend. However, not all of these isolates indicate genuine lung disease [38]. No matter the explanation, reported data all seem to suggest an overall increasing trend of NTM for both infection and disease. However, the study has several limitations. Because data from a longer period were not available for many countries, we were only able to include a few studies from Eastern European and South American countries and no studies from Africa at all. Interestingly, a study by Hermansen et al. did not prove any significant trend for either infection or disease despite being a nationwide study reporting NTM over the longest time span, challenging the claim of an increasing trend [39]. Our material covered a period of 10 years. The last two years were more difficult to interpret due to the Covid-19 pandemic, lower reporting, and additionally the occurrence of the war in Ukraine.

According to WHO estimates, among the 10 million TB patients in 2017, 5.8 million were men, 3.2 million were women, and 1 million were children. In Poland, there is also male dominance among TB patients, particularly in the 45–64 age group. The gender distribution in NTM-LD is different. In our study, the gender distribution is almost equal. But in other studies, however, males dominated [40,41].

MAC and *M. kansasii* were the most common organisms causing disease. On numerous Japanese materials, the most common pathogenic organism among NTM-LD cases was MAC (n = 1881; 87.3%), followed by *M. abscessus* complex (MABC) (n = 119; 5.5%), *M. kansasii* (n = 83; 3.9%), and *M. fortuitum* (n = 28; 1.3%) [42]. In our study, in contrast to Japanese material, the most common pathogenic organism among NTM-LD cases was *M. kansasii*, followed by MAC. Other studies show a similar trend [20,41,43]. It is presumably uncommon in Northern Israel. Most cases attributed to *M. kansasii* infection had underlying disease. A survey from Israel of *M. kansasii* between 1998 and 2004 showed no association with HIV and high rates of associated lung disease [38]. In our material, the significant predominance of *M. kansasii* and MAC is confirmed. This predominance is consistent with previous studies from Poland, Denmark, and the United States [18,44,45].

NTM diseases may be established by chest radiography performed for screening purposes. However, the radiological findings of NTM-LD may vary depending on the causative organism [46,47,48]. Findings that are consistent with NTM-LD on chest X-ray or high-resolution computed tomography (CT) include infiltration (usually nodular or reticular nodular), cavity, multifocal bronchiectasis, and/or multiple nodules [22]. In our material, we identified over the years there are more often cases of mycobacteriosis with caves and involvement of both lungs. In comparison, there were about 20% more patients with caves in their lungs in 2022 than in 2013. The lesions, which show more cavities after 2021, show that, due to the pandemic, delayed diagnosis has resulted in a higher percentage of patients with bilateral lesions at the expense of previously observed unilateral infiltrative lesions.

Characteristic imaging findings suggest that pulmonary MAC diseases include the following two types: nodular/bronchiectatic and fibrocavitary diseases, with the former considered to account for the majority of cases, being commonly observed among middle-aged and older women without a history of smoking.

Looking further into our study, we further noticed limitations. Firstly, it is difficult to access the clinical data of the patients studied. Caution should be exercised in assuming that the NTM isolation rate does not reflect the clinical infection rate. Therefore, results should be interpreted with caution. Secondly, some NTM species may be missed by the GenoType Mycobacterium tests for identifying NTM species. Thirdly, a small number of extrapulmonary samples were included in our study. There is another health center for HIV patients in Bydgoszcz, which is why there are so few patients in our study. The last three years of the research were at the same time as the pandemic of COVID-19. This should be considered as a reason for the decrease in the number of specimens required for mycobacterial culture, and the consequent decrease in the numbers of NTM strains and NTM-LD cases detected. Therefore, the results may not be reliable.

In summary, our study, despite these limitations, contributed to a better understanding of the dynamics of NTM infections in Poland. From 2013 to 2022 about 395 NTM isolations were identified, which allowed us to classify 149 of the cases as “definite NTM disease”. The rest of the patients with cultured NTM were considered as colonized patients or the source of NTM strains could be contaminated from medical devices such as bronchoscope. Our observations are in keeping with some reports by others [1,49,50]. We highlight the need for the establishment of a systematic approach to diagnose NTM disease and uniform reporting to assess the real NTM disease epidemiology. Overall, in Europe, information on the burden of NTM is fragmentary, and available information and estimates vary greatly between countries. This article presents an important addition to the available knowledge about NTM-LD by using administrative data linked to clinical data to estimate burden, cost, and mortality, adjusted for comorbid conditions.

In conclusion, *M. kansasii* and MAC are the most prevalent NTM isolates among respiratory samples in Northern Poland. In addition, an increase in MAC cultivation and a decrease in *M. kansasii* cultivation in the last years were reported. Despite unforeseen episodes in recent years, such as the COVID-19 pandemic and the war in Ukraine, the results of our study indicate that the annual prevalence of NTM-LD remains stable. However, identifying significant trends requires further intensive research.

## Figures and Tables

**Figure 1 pathogens-12-00988-f001:**
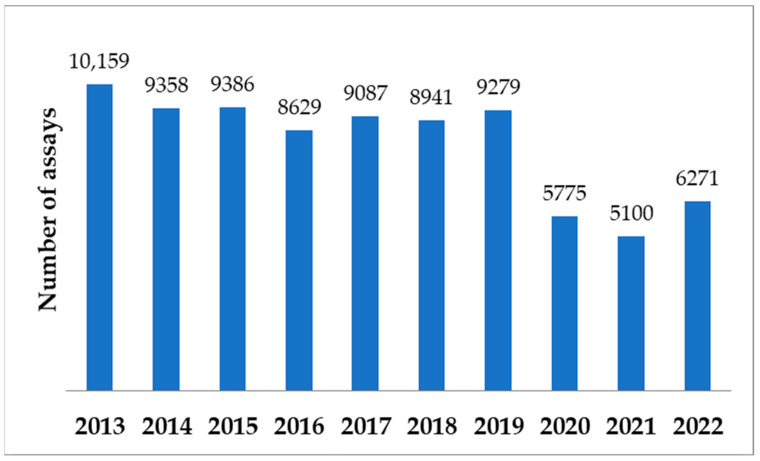
Numbers of specimens submitted for mycobacterial culture during the 10-year analysis period.

**Figure 2 pathogens-12-00988-f002:**
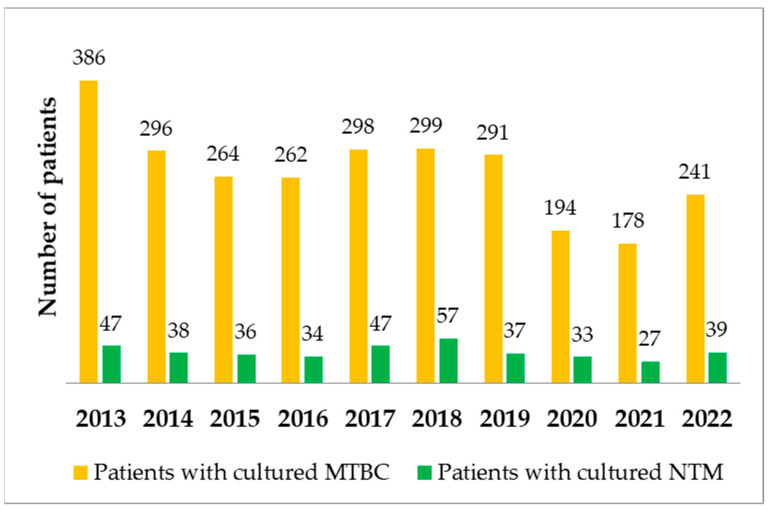
Numbers of patients with MTBC and NTM strains during the 10-year analysis period.

**Figure 3 pathogens-12-00988-f003:**
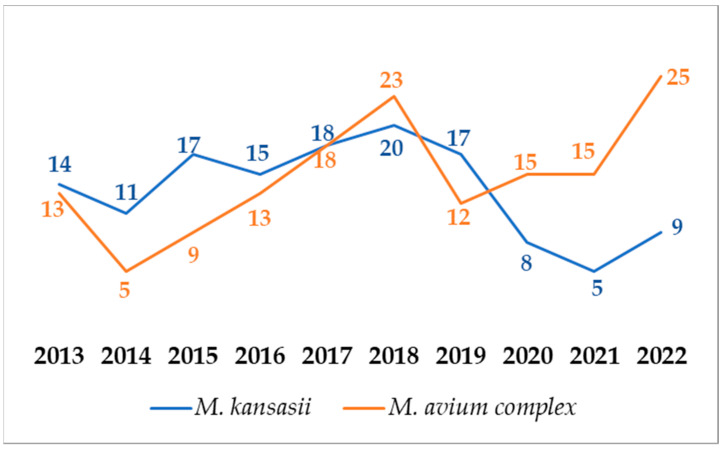
Numbers MAC and *M. kansasii* strains during the 10-year analysis period.

**Figure 4 pathogens-12-00988-f004:**
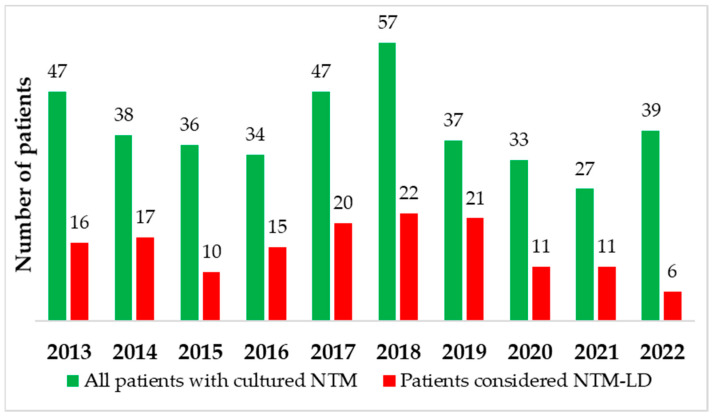
Numbers of patients with NTM-LD among all patients with cultured NTM during the 10-year analysis period.

**Figure 5 pathogens-12-00988-f005:**
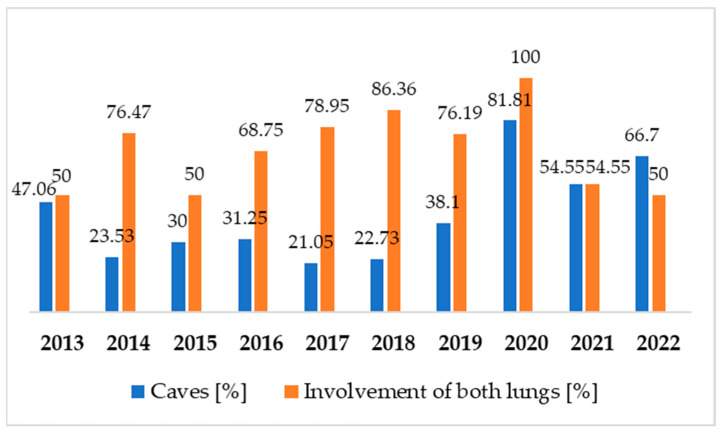
Radiological image: presence of caves and involvement of both lungs over the years.

**Table 1 pathogens-12-00988-t001:** NTM species isolated from clinical specimens between 2013 and 2022.

NTM Species	2013	2014	2015	2016	2017	2018	2019	2020	2021	2022	Total
*M. abscessus*	1	0	0	0	0	0	0	1	0	0	2
*M. avium*	9	4	9	9	18	17	10	12	11	18	117
*M. chelonae*	0	0	1	0	2	0	0	0	0	0	3
*M. chimaera*	0	0	0	0	0	1	1	1	2	4	9
*M. fortuitum*	2	3	2	1	0	1	0	0	2	0	11
*M. gordonae*	8	11	2	1	5	4	3	1	1	3	39
*M. interjectum*	0	0	0	0	0	1	0	0	0	0	1
*M. intracellulare*	4	1	0	4	0	5	1	2	2	3	22
*M. kansasii*	14	11	17	15	18	20	17	8	5	9	134
*M. lentiflavum*	0	2	0	0	0	0	0	0	0	0	2
*M. mageritense*	0	0	0	0	0	1	0	0	0	0	1
*M. malmoense*	0	2	0	1	0	1	1	1	0	1	7
*M. neoaurum*	0	0	1	0	0	0	0	0	0	0	1
*M. nonchromogenicum*	0	0	1	0	0	0	0	0	0	0	1
*M. peregrinum*	1	0	0	1	0	0	1	0	0	0	3
*M. szulgai*	0	1	0	0	0	0	0	0	1	1	3
*M. xenopi*	5	2	1	1	2	2	3	3	2	0	21
other *RGM*	3	1	1	1	2	4	0	4	1	0	17
Total	47	38	36	34	47	57	37	33	27	39	395

**Table 2 pathogens-12-00988-t002:** NTM species among patients with NTM-LD.

NTM Species	2013	2014	2015	2016	2017	2018	2019	2020	2021	2022	Total
*M. kansasii*	9	10	8	9	12	12	10	4	3	0	77
*MAC*	0	4	0	5	7	9	9	3	5	4	46
*M. xenopi*	4	1	1	0	1	0	1	3	2	0	13
*M. malmoense*	0	1	0	1	0	0	1	1	0	1	5
*M. gordonae*	2	0	1	0	0	0	0	0	0	0	3
*M. szulgai*	0	0	0	0	0	0	0	0	1	1	2
*M. abscessus*	1	0	0	0	0	0	0	0	0	0	1
*M. fortuitum*	0	1	0	0	0	0	0	0	0	0	1
*M. interjectum*	0	0	0	0	0	1	0	0	0	0	1
Total	16	17	10	15	20	22	21	11	11	6	149

**Table 3 pathogens-12-00988-t003:** Clinical characteristics in NTM-LD patients.

Underlying Condition	n	(%)
Lung disease	41	(27.5)
Cancer	10	(6.7)
Diabetes	11	(7.4)
Immunodeficiency:		
Hematological malignancies	3	(2.0)
Biological treatment	2	(1.3)
HIV	2	(1.3)

## Data Availability

Data supporting reported results can be found in source data collected in the Regional Center of Pulmonology in Bydgoszcz, Poland.

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
