# Peer review of "Trends from the Last Decade with Nontuberculous Mycobacteria Lung Disease (NTM-LD): Clinicians’ Perspectives in Regional Center of Pulmonology in Bydgoszcz, Poland"

_pathogens, 2023, doi:10.3390/pathogens12080988_

Round 1

Reviewer 1 Report

The manuscript presents a large material covering a ten-year observation of MOTT infections in one region of Poland. The lack of similar works from Poland justifies its publication even more. However, it is necessary to make major changes because the work is badly written. Too much unnecessary repetition in the introduction and in discussion. Methodology too briefly described. The results require greater precision and deeper presentation. Too lengthy discussion should be shortened, threads inadequate to the results obtained should be removed and it should be enriched with actual discussions of very interesting results. The presentation of the literature needs to be unified.

Abstract

The summary is very badly written and needs a thorough correction. It is necessary to unify the data, because different data are in the text of the article and different in the abstract. This concerns, for example, the presence of M. kansasi in 77.52% and M. avium in 46.31%, which does not correspond to the results in Figure 6. The authors in the introduction to the abstract mentions 125 NTM species but in the paper 180. The background does not explain why these studies were undertaken, and the goal is too vague. Age characteristics from the minimum to the maximum does not show what kind of patients they were, the average age or median is necessary to be able to better assess this group. Numerous unnecessary repetitions must be removed from the summary (e.g. 149 cases are mentioned three times). HIV is not a disease but a pathogen. It is necessary to clearly distinguish NTM from NTM-LD, which the authors do not always do in the abstract and it may cause misunderstanding of the problem. The assessment of patients was not limited to clinical and microbiological assessment, but was much broader, including also radiological assessment. This was not mentioned in the summary. Conclusions need to be redefined, are too vague and do not follow fully from the results. On what basis the authors claim that NTM-LD are frequent, because we do not know how large a group they constituted among hospital. In the abstract, the authors showed a different location of the changes, and in the conclusions, there are different clinical manifestations? I propose to rewrite the abstract completely, since I have only pointed out the major shortcomings.

Introduction

The introduction requires correction, because is characterized by unnecessary repetitions and even three entries of the same statements, e.g.:

how many mycobacteria are pathogenic [39-40 and 72-73],

increase in TMN diseases associated with improved breeding techniques and physician awareness [48-50, 55-56, 84-88-89],

dominance of NTM in the lungs [56-57, 60-67, 70-71],

extrapulmonary locations [44-45, 68-69],

the epidemiology of NTM diseases indicating an increase in cases is too long-winded in several places [47-48, 58-62, 66-68],

occurrence of NTM diseases in immune-compromised patients [52-53, 55-56, 75-77, 89-90].

I have doubts whether diabetes and COPD are immune-modulatory diseases and I propose to define it differently [53-54].

Data on Poland are not cited and the data is not clearly defined [64-65]. In Poland, there is a theoretical obligation to register NTM diseases, but it is not implemented.

The authors do not indicate which epidemiological data in individual countries are the result of a local analysis of the epidemiological situation and which result from the mandatory nationwide registration of each case.

Some specific information requires an indication of its source, or the articles provided are inadequate to the matters discussed, because they are not the primary and secondary source of information [e.g. 42-46, 79-81, 91-94]. It is advisable to supplement references.

The term NTM disease is used instead of NTM diseases [47-48, 55-65].

Some citations are inadequate to the issues discussed because they refer to the local situation and are transferred to the general situation [e.g. 19, 21].

Is MOTT found only in aerosols or also in the air [74]?

Has the improvement of breeding techniques and the awareness of doctors increased the incidence or rather detection of these diseases?

It is unclear in lines 91-92 whether nodules and cavities promote NTM diseases or are they a typical change in its course?

Citation 25 should end the sentence on line 102.

Bad spaces on line 77.

The sentence in lines 81-82 needs rewording.

The authors do not mention the different evaluation of sputum and bronchoalveolar lavage fluid cultures.

There is no mention of MOTT contamination of equipment used in the diagnosis of the disease, which causes false positive results not due to colonization of the organism.

I suggest significantly shortening the introduction by eliminating unnecessary repetitions and clarifying unclear fragments by supplementing the references.

Methods

Bacteriological methodology is briefly described, although it schuld contain references from the literature. However, there is no description of the methodology of clinical and radiological assessment, which should be supplemented.

In the methodology, the materials from which the mycobacteria were isolated were calculated and the results were repeated specifying the numbers. Figure 1 shows a steady number of assays and a significant decrease from 2020 to 2022 during the COVID19 pandemic. The description in the text suggests a more gradual decline in the number of cultures over a decade [137].

The methodology does not specify the criteria for the diagnosis of tuberculosis and MOTT. It has not been confirmed to what extent all cases met the ATS criteria and to what extent ATS/ERS/CDC. The methodology did not specify what the radiological diagnosis was based on, whether it was only changes in the chest X-ray or computed tomography of the lungs, or maybe both methods at the same time. The methodology for assessing the clinical changes that were analyzed was not described.

Results

We do not know how many cases were diagnosed with tuberculosis out of 2709 patients with cultured MTBC. In the case of tuberculosis, the number of patients could be higher because 20-40% of patients with tuberculosis have no bacteriological confirmation.

It should be clarified what is shown in Figure 2, whether patients with NTM-LD and patients with tuberculosis, or the number of positive cultures, which in the case of MOTT were largely colonization and not disease.

The captions under the figures are misleading because we are talking about both patients and isolates and it is not the same.

It is important to describe on what basis people with MOTT cultures were not classified as people with mycobacteriosis.

We dont know how many culture-negative tuberculosis patients there were?

The caption under Figure 3 shows the number of specimens with NTM strains, but it also shows the percentages.

It should be noted that RMG was cultured in 38 cases with the percentage of the total (9.6%).

In fact, it should be written that MAC was bred most often and M. kansasi in second place.

One should not write about the decrease in the detection of M. kansasi, but rather about the less frequent cultivation of this bacterium.

Detection methods have improved and the number of cases reported has decreased. This is not consistent with the entry in the introduction and discussion.

The first sentence in section 3.4 Clinical significance should be completely redrafted.

It is necessary to specify what lung diseases were found in 41 patients with NTM-LD. This is very important for the interpretation of the frequency of symptoms or radiological changes discussed later.

It is necessary to define what immunodeficiencies were found in 7 cases.

Table I should be presented differently because the entry is misleading and may mean that women were diagnosed with cancer and diabetes, while men suffered from immune disorders.

It is not clear why 2 patients with 4 HIV(+) were excluded from the analyzed group. The fact that they died before treatment or that they disappeared from follow-up does not change the fact that they were with mycobacteriosis if the criteria were met.

We do not know what kind of coexisting cancer is mentioned in 10 patients. Was it lung cancer or something else? Were these patients treated with chemo- or radiotherapy, were there cases of haematological malignancies among them? These patients could then be included in the immunocompromised group.

It is very important to know whether patients with cough did not have other lung diseases, e.g. bronchiectasis, asthma or COPD. It is then difficult to determine whether the cough resulted from mycobacteriosis or from a concomitant disease.

It is not known how long the cough lasted or was it productive? In the case of patients with haemoptysis is identical. We need to know to the extent that it was found in patients with other lung diseases, e.g. with bronchiectasis.

We do not know which group of patients had fever. How many of them had comorbidities, cancer or immune disorders?

In this section, the authors introduced the term mycobacteriosis previously used NTM diseases. A standardized nomenclature should be used.

Lack of BMI means that it is not known to what extent a decrease in BMI was observed in this group?

The results lack the determination of the frequency of nodular lesions and the determination of the areas of the lungs where radiological changes were found.

For unknown reasons, the authors did not discuss how NTM-LD patients were treated - with what effect and for how long. How the treatment affected the resolution of radiological changes and clinical symptoms.

There is also no assessment of the impact of individual factors (e.g. age, sex, comorbidities) on the clinical picture, on the occurrence and course of the disease.

The radiological picture is not discussed according to the type of bacteria. Are there differences between MAC and M. kansasi infection?

Certainly, the last point of Trends should be completely abandoned, because it is a literal repetition of the information contained in the text and illustrated by figures. No need to write about it a third time.

Discussion

The discussion is badly written because it is inadequate to the results obtained. Requires major changes. It is incomprehensible to promote the thesis about the increase in the incidence of mycobacteriosis and to quote works from the literature, while the presented results do not show it. Perhaps this is due to the presentation of positive breeding results and not cases of mycobacteriosis, which do not differ significantly in the last 10 years. There was indeed an increase in cultured MACs in the last 3 years, but was it an increase in diseases? The increase in infections on MAC is accompanied by a decrease in the detection of M. kansasi. It is now a fairly common phenomenon of replacing M. kansasi, which used to dominate in Poland and Europe, in favor of MAC. The trends observed in Poland with an increase in MAC and a decrease in M. kansasi are therefore not surprising and are observed throughout Europe.

The changes observed during the COVID19 pandemic may result from the impact of the pandemic on the detection and occurrence of infectious diseases. In general, a significant decrease in registered infectious diseases, including tuberculosis, was observed at that time around the world. It is worth discussing this in more detail in the discussion.

However, we do not know how the incidence was in individual years because there is a significant difference between the results from breeding (395) and patients with mycobacteriosis (149 cases). Therefore, it is worth presenting the distribution of diagnosed cases each year in the results to be able to talk about the increase in NTM-LD cases.

Too much space in the discussion was devoted to epidemiological problems, while the work did not concern only this aspect. Information on this subject is quite chaotically presented, not always consistent with the data in the introduction and with the results. I suggest shortening and organizing this thread.

The increase in the incidence of mycobacteriosis in the world is clearly influenced by the spread of HIV infections. In Poland, the number of these infections is much lower than in many other countries whose data were quoted in the paper. This may result in a different epidemiology in Poland compared to other countries. This is worth wrote it in the discussion. It is this phenomenon that may cause different epidemiological trends in Poland.

It is worth noting in the discussion that the number of men and women suffering from NTM-LD is similar (54%/46%), while in the case of tuberculosis, men suffer twice as often.

The impact of comorbidities on the occurrence and course of NTM-LD should be discussed. For this purpose, it would be necessary to better characterize these accompanying diseases, because we do not even know what diseases they were.

The authors introduced the abbreviation NTM PD not used before (???).

There is no citation for MOTT registration in Denmark.

The term increase for both infection and disease is not understood [281 and 323]. Authors must be very consistent in using the appropriate nomenclature to distinguish mycobacteriosis from colonization or environmental contamination. They do not always clearly differentiate these concepts and it is necessary to review the text in this respect and clarify what is mentioned in a given fragment of the work.

In the light of the presented results, the sentence The majoraty of NTM pulmonary isolates in our institution represent disease [288-289] is incorrect.

The authors did not discuss another topic that affects the epidemiological data on tuberculosis and possibly also MOTT infections. It is about the war in Ukraine and the ensuing mass influx of immigrants from areas with a much worse epidemiological situation than in Poland.

The discussion raised the topic of differences in the radiological picture depending on the pathogens, but the sources of this information were not indicated. It is necessary to supplement the literature in this paragraph. It is not shown in the results whether such differences were observed, which is a pity. It's worth supplementing. This part is much more extensive than the presentation of radiological results. The discussion does not refer to the presented results, and it should.

In the next part, the authors return to the thesis of the increase in the incidence of NTM-LD, which has not been demonstrated [323-324]. The record of data analysis on NTM infections is unclear [328-333]. The paper does not present the results of this analysis and it is not clear what this part is about. Interestingly, Hermansen's results, contrary to the record in the discussion, are consistent with the results presented in this paper.

The postulate about the need to register each case of NTM-LD is not understandable. It is not contagious and has no social significance. Of course, it would be helpful to monitor the scale of the problem, but this postulate is not justified in relation to many other more important health problems, where such mandatory registration would be really helpful and necessary.

On what basis did the authors conclude that the cases of NTM-LD were not dependent on population density?

The end of the discussion is too general, too much emphasis is placed on epidemiology and little space is devoted to clinical aspects. It does not indicate which patients require diagnostics towards NTM-LD, what is the effectiveness of diagnostic and therapeutic methods. It is worth expanding these aspects in the discussion.

What is NB disease [319]?

References

References requires profound changes.

Each item is written in a different style. It is necessary to standardize the notation in accordance with the instructions for MDPI authors:

Author 1, A.B.; Author 2, C.D. Title of the article. Abbreviated Journal Name Year, Volume, page range.

The authors use a lot of freedom in writing abbreviations of letters and monograms of names, using dots or not. It is similar with the abbreviations of magazines, sometimes with a dot, sometimes without, and sometimes only at the end of the magazine. It needs to be unified.

Sometimes the volume is in bold, sometimes the year is in bold, and sometimes all numbers are in normal or all numbers are in bold.

Sometimes the name of the authors was entered in full, e.g. poem 444 item 17.

Sometimes there are no spaces between words and often unnecessary numerous spaces, e.g. between monograms of names.

Item 2 was reviewed before the study was undertaken.

Some articles link to Pubmed or Goegel Scholar, while others do not, although they are available.

The number of authors before et al. is very variable. from one, by 3 authors, 6 authors. Sometimes more than 6 authors without introducing the abbreviation et al. It needs to be unified.

In position 43 commas before the surname and not after the monograms. '

Some references use & before the last author, but most are preceded by a comma.

In some articles, all words are capitalized and in others, only the first word in a sentence is capitalized.

It is worth correcting the text from the linguistic side, because it is inaccurate in places and contains language errors. It is also worth using a native speaker to improve the text.

Author Response

We thank the reviewers for their careful review of our manuscript entitled “Trends from the last decade with Nontuberculous Mycobacteria Lung Disease (NTM-LD): clinicians’ perspectives in Regional Center of Pulmonology in Bydgoszcz, Poland”. We have carefully read the comments of the reviewers and have revised our manuscript in light of them.

All co-authors of our manuscript are aware of and agree with the revisions. Detailed point-by-point responses to the reviewers’ comments are provided below.

Thank you once again for considering our manuscript. We look forward to hearing from you regarding our submission.

Point-by-point response to reviewers

1.Faced with suggestions for linguistic improvement. Of course, we completely agree with this comment. The article was consulted by an English specialist, but we concluded that we would ask for an improvement in the language form of the MDPI.

2.The number of key words was increased.

3.As suggested by the reviewers, the entire content of the manuscript was re-examined and a strongly revised version was submitted for further analysis.

4.Due to the rewriting of the manuscript, the numbering of figures and tables has changed

  1. Figure 4 has been replaced by Table 1. We believe that the table significantly improves the way these data are shown
  2. The previous Table 1 (after the change – Table 2) has been reworded. The Table 2 shows underlying conditions, while demographic conditions are described in the results.
  3. Lines numbered in the original will have a different numbering in the revison manuscript.

The manuscript presents a large material covering a ten-year observation of MOTT infections in one region of Poland. The lack of similar works from Poland justifies its publication even more. However, it is necessary to make major changes because the work is badly written. Too much unnecessary repetition in the introduction and in discussion. Methodology too briefly described. The results require greater precision and deeper presentation. Too lengthy discussion should be shortened, threads inadequate to the results obtained should be removed and it should be enriched with actual discussions of very interesting results. The presentation of the literature needs to be unified.

Abstract

The summary is very badly written and needs a thorough correction. It is necessary to unify the data, because different data are in the text of the article and different in the abstract. This concerns, for example, the presence of M. kansasi in 77.52% and M. avium in 46.31%, which does not correspond to the results in Figure 6. The authors in the introduction to the abstract mentions 125 NTM species but in the paper 180. The background does not explain why these studies were undertaken, and the goal is too vague. Age characteristics from the minimum to the maximum does not show what kind of patients they were, the average age or median is necessary to be able to better assess this group. Numerous unnecessary repetitions must be removed from the summary (e.g. 149 cases are mentioned three times). HIV is not a disease but a pathogen. It is necessary to clearly distinguish NTM from NTM-LD, which the authors do not always do in the abstract and it may cause misunderstanding of the problem. The assessment of patients was not limited to clinical and microbiological assessment, but was much broader, including also radiological assessment. This was not mentioned in the summary. Conclusions need to be redefined, are too vague and do not follow fully from the results. On what basis the authors claim that NTM-LD are frequent, because we do not know how large a group they constituted among hospital. In the abstract, the authors showed a different location of the changes, and in the conclusions, there are different clinical manifestations? I propose to rewrite the abstract completely, since I have only pointed out the major shortcomings.

The abstract has been throughly rewritten in accordance with the insights provided in the commentary. Incorrect numbers and percentages of mycobacteria, resulting from a mistake, have been corrected. The number of NTM species has been corrected, in accordance with the data included in the introduction. The repetitions have been removed. The median age has been added. The radiological assessment information has been added. 

Introduction

The introduction requires correction, because is characterized by unnecessary repetitions and even three entries of the same statements, e.g.:

how many mycobacteria are pathogenic [39-40 and 72-73],

The repetition has been removed.

increase in TMN diseases associated with improved breeding techniques and physician awareness [48-50, 55-56, 84-88-89],

The repetition has been removed.

dominance of NTM in the lungs [56-57, 60-67, 70-71],

The repetition has been removed.

extrapulmonary locations [44-45, 68-69],

The repetition has been removed.

the epidemiology of NTM diseases indicating an increase in cases is too long-winded in several places [47-48, 58-62, 66-68],

The repetition has been removed.

occurrence of NTM diseases in immune-compromised patients [52-53, 55-56, 75-77, 89-90].

The repetition has been removed.

I have doubts whether diabetes and COPD are immune-modulatory diseases and I propose to define it differently [53-54].

It has been rewritten.

Data on Poland are not cited and the data is not clearly defined [64-65]. In Poland, there is a theoretical obligation to register NTM diseases, but it is not implemented.

The authors do not indicate which epidemiological data in individual countries are the result of a local analysis of the epidemiological situation and which result from the mandatory nationwide registration of each case.

Some specific information requires an indication of its source, or the articles provided are inadequate to the matters discussed, because they are not the primary and secondary source of information [e.g. 42-46, 79-81, 91-94]. It is advisable to supplement references.

Following a suggestion from one of the reviewers that the introduction was too long, it has been shortened and some elements have been reworded to harmonise the whole chapter.

The term NTM disease is used instead of NTM diseases [47-48, 55-65].

As suggested by the reviewer, improved

Some citations are inadequate to the issues discussed because they refer to the local situation and are transferred to the general situation [e.g. 19, 21].

Following a suggestion from one of the reviewers that the introduction was too long, it has been shortened and some elements have been reworded to harmonise the whole chapter.

Is MOTT found only in aerosols or also in the air [74]?

Following a suggestion from one of the reviewers that the introduction was too long, it has been shortened and some elements have been reworded to harmonise the whole chapter.

Has the improvement of breeding techniques and the awareness of doctors increased the incidence or rather detection of these diseases?

As suggested by the reviewer, improved.

It is unclear in lines 91-92 whether nodules and cavities promote NTM diseases or are they a typical change in its course?

Following a suggestion from one of the reviewers that the introduction was too long, it has been shortened and some elements have been reworded to harmonise the whole chapter.

Citation 25 should end the sentence on line 102.

Following a suggestion, it has been fixed.

Bad spaces on line 77.

It has been fixed.

The sentence in lines 81-82 needs rewording.

The authors do not mention the different evaluation of sputum and bronchoalveolar lavage fluid cultures.

It has been clarified in Materials and methods section.

There is no mention of MOTT contamination of equipment used in the diagnosis of the disease, which causes false positive results not due to colonization of the organism.

It has been reworded. The consideration about MOTT contaminations has been added to introduction and discussion sections.

I suggest significantly shortening the introduction by eliminating unnecessary repetitions and clarifying unclear fragments by supplementing the references.

Following a suggestion from one of the reviewers that the introduction was too long, it has been shortened and some elements have been reworded to harmonise the whole chapter.

Methods

Bacteriological methodology is briefly described, although it schuld contain references from the literature.

In fact, there is a reference – “The methods used in Department of Microbiology were in accordance with the methodology in force in mycobacteria laboratories in Poland [27]”.

However, there is no description of the methodology of clinical and radiological assessment, which should be supplemented.

As suggested by the reviewer, the methodology for clinical and radiological assessment, is completed.

In the methodology, the materials from which the mycobacteria were isolated were calculated and the results were repeated specifying the numbers. Figure 1 shows a steady number of assays and a significant decrease from 2020 to 2022 during the COVID19 pandemic. The description in the text suggests a more gradual decline in the number of cultures over a decade [137].

The description in the text has been rewritten.

The methodology does not specify the criteria for the diagnosis of tuberculosis and MOTT. It has not been confirmed to what extent all cases met the ATS criteria and to what extent ATS/ERS/CDC. The methodology did not specify what the radiological diagnosis was based on, whether it was only changes in the chest X-ray or computed tomography of the lungs, or maybe both methods at the same time. The methodology for assessing the clinical changes that were analyzed was not described.

As suggested by the reviewer, the methodological part of the scope has been completely revised

Results

We do not know how many cases were diagnosed with tuberculosis out of 2709 patients with cultured MTBC. In the case of tuberculosis, the number of patients could be higher because 20-40% of patients with tuberculosis have no bacteriological confirmation.

The sentence has been edited. All of the patients with cultured MTBC were considered as tuberculosis case. We have not mentioned about number of patients with bacteriologically unconfirmed tuberculosis because it is not the purpose of this work.

It should be clarified what is shown in Figure 2, whether patients with NTM-LD and patients with tuberculosis, or the number of positive cultures, which in the case of MOTT were largely colonization and not disease.

It has been rewritten. We have clarified Figure 2. It shows patients with cultured NTM and cultured MTBC.

The captions under the figures are misleading because we are talking about both patients and isolates and it is not the same.

It has been corrected.

It is important to describe on what basis people with MOTT cultures were not classified as people with mycobacteriosis.

It was clarified. “All of the 395 patients with NTM strains were evaluated. Clinical and radiographic criteria of NTM-LD were met in 149 case, every case was considered “definite”. The rest of cases (n = 246) did not meet the criteria of NTM-LD according to ATS.”

We dont know how many culture-negative tuberculosis patients there were?

We have not mentioned about number of patients with bacteriologically unconfirmed tuberculosis because it is not the purpose of this work.

The caption under Figure 3 shows the number of specimens with NTM strains, but it also shows the percentages.

It should be noted that RMG was cultured in 38 cases with the percentage of the total (9.6%).

Figure 4 has been replaced by Table 1. We believe that the table significantly improves the way theses data are shown.

In fact, it should be written that MAC was bred most often and M. kansasi in second place.

We focus on species analysis in this section. M. kansasii was the most frequently cultured species, followed by M. avium. MAC complex is a group that consists of M. avium, M. intracellulare and M. chimaera. Numbers of MAC complex and M. kansasii during the 10-year analysis period were compared later.

One should not write about the decrease in the detection of M. kansasi, but rather about the less frequent cultivation of this bacterium.

It has been rewritten.

Detection methods have improved and the number of cases reported has decreased. This is not consistent with the entry in the introduction and discussion.

This sentence has been rewritten, both in the introduction and discussion.

The first sentence in section 3.4 Clinical significance should be completely redrafted.

The first sentence in section 3.4 was redrafted.

It is necessary to specify what lung diseases were found in 41 patients with NTM-LD. This is very important for the interpretation of the frequency of symptoms or radiological changes discussed later.

It is now written, what kind of lung diseases were found in 41 patients with NTM-LD.

It is necessary to define what immunodeficiencies were found in 7 cases.

The immunodeficiencies have been defined.

Table I should be presented differently because the entry is misleading and may mean that women were diagnosed with cancer and diabetes, while men suffered from immune disorders.

The previous Table 1 (after the change – Table 2) has been reworded. The Table 2 shows underlying conditions, while demographic conditions are described in the results.

It is not clear why 2 patients with 4 HIV(+) were excluded from the analyzed group. The fact that they died before treatment or that they disappeared from follow-up does not change the fact that they were with mycobacteriosis if the criteria were met.

This point has been explained more thoroughly.

We do not know what kind of coexisting cancer is mentioned in 10 patients. Was it lung cancer or something else? Were these patients treated with chemo- or radiotherapy, were there cases of haematological malignancies among them? These patients could then be included in the immunocompromised group.

It is now written, what kind of cancer suffered from patients in oncological group. Cases of haematological malignancies were included in immunocompromised group from the beginning. 

It is very important to know whether patients with cough did not have other lung diseases, e.g. bronchiectasis, asthma or COPD. It is then difficult to determine whether the cough resulted from mycobacteriosis or from a concomitant disease.

Patients with other lung diseases has been mentioned in the group with a cough.

It is not known how long the cough lasted or was it productive? In the case of patients with haemoptysis is identical. We need to know to the extent that it was found in patients with other lung diseases, e.g. with bronchiectasis.

It has been rewritten.

We do not know which group of patients had fever. How many of them had comorbidities, cancer or immune disorders?

Specific data has been added.

In this section, the authors introduced the term mycobacteriosis previously used NTM diseases. A standardized nomenclature should be used.

As suggested by the reviewers, this paragraph has been corrected

Lack of BMI means that it is not known to what extent a decrease in BMI was observed in this group?

In terms of this element, the topic of the article was not expanded upon.Sadly, the research does not include changes in BMI.

The results lack the determination of the frequency of nodular lesions and the determination of the areas of the lungs where radiological changes were found.

Specific data about nodular lesions were added. 

For unknown reasons, the authors did not discuss how NTM-LD patients were treated - with what effect and for how long. How the treatment affected the resolution of radiological changes and clinical symptoms.

It is added information about the tolerance, result and duration of the treatments.

There is also no assessment of the impact of individual factors (e.g. age, sex, comorbidities) on the clinical picture, on the occurrence and course of the disease.

It described the impact of age and sex on the presence of multiple symptoms.

The radiological picture is not discussed according to the type of bacteria. Are there differences between MAC and M. kansasi infection?

A paragraph about radiological differences between MAC and M. kansasii infection is added.

Certainly, the last point of Trends should be completely abandoned, because it is a literal repetition of the information contained in the text and illustrated by figures. No need to write about it a third time.

As suggested by the reviewers, the deletion of this paragraph has been made.

Discussion

The discussion is badly written because it is inadequate to the results obtained. Requires major changes. It is incomprehensible to promote the thesis about the increase in the incidence of mycobacteriosis and to quote works from the literature, while the presented results do not show it. Perhaps this is due to the presentation of positive breeding results and not cases of mycobacteriosis, which do not differ significantly in the last 10 years. There was indeed an increase in cultured MACs in the last 3 years, but was it an increase in diseases? The increase in infections on MAC is accompanied by a decrease in the detection of M. kansasi. It is now a fairly common phenomenon of replacing M. kansasi, which used to dominate in Poland and Europe, in favor of MAC. The trends observed in Poland with an increase in MAC and a decrease in M. kansasi are therefore not surprising and are observed throughout Europe.

The changes observed during the COVID19 pandemic may result from the impact of the pandemic on the detection and occurrence of infectious diseases. In general, a significant decrease in registered infectious diseases, including tuberculosis, was observed at that time around the world. It is worth discussing this in more detail in the discussion.

However, we do not know how the incidence was in individual years because there is a significant difference between the results from breeding (395) and patients with mycobacteriosis (149 cases). Therefore, it is worth presenting the distribution of diagnosed cases each year in the results to be able to talk about the increase in NTM-LD cases.

Too much space in the discussion was devoted to epidemiological problems, while the work did not concern only this aspect. Information on this subject is quite chaotically presented, not always consistent with the data in the introduction and with the results. I suggest shortening and organizing this thread.

The increase in the incidence of mycobacteriosis in the world is clearly influenced by the spread of HIV infections. In Poland, the number of these infections is much lower than in many other countries whose data were quoted in the paper. This may result in a different epidemiology in Poland compared to other countries. This is worth wrote it in the discussion. It is this phenomenon that may cause different epidemiological trends in Poland.

It is worth noting in the discussion that the number of men and women suffering from NTM-LD is similar (54%/46%), while in the case of tuberculosis, men suffer twice as often.

The impact of comorbidities on the occurrence and course of NTM-LD should be discussed. For this purpose, it would be necessary to better characterize these accompanying diseases, because we do not even know what diseases they were.

The authors introduced the abbreviation NTM PD not used before (???).

There is no citation for MOTT registration in Denmark.

The term increase for both infection and disease is not understood [281 and 323]. Authors must be very consistent in using the appropriate nomenclature to distinguish mycobacteriosis from colonization or environmental contamination. They do not always clearly differentiate these concepts and it is necessary to review the text in this respect and clarify what is mentioned in a given fragment of the work.

In the light of the presented results, the sentence The majoraty of NTM pulmonary isolates in our institution represent disease [288-289] is incorrect.

The authors did not discuss another topic that affects the epidemiological data on tuberculosis and possibly also MOTT infections. It is about the war in Ukraine and the ensuing mass influx of immigrants from areas with a much worse epidemiological situation than in Poland.

The discussion raised the topic of differences in the radiological picture depending on the pathogens, but the sources of this information were not indicated. It is necessary to supplement the literature in this paragraph. It is not shown in the results whether such differences were observed, which is a pity. It's worth supplementing. This part is much more extensive than the presentation of radiological results. The discussion does not refer to the presented results, and it should.

In the next part, the authors return to the thesis of the increase in the incidence of NTM-LD, which has not been demonstrated [323-324]. The record of data analysis on NTM infections is unclear [328-333]. The paper does not present the results of this analysis and it is not clear what this part is about. Interestingly, Hermansen's results, contrary to the record in the discussion, are consistent with the results presented in this paper.

The postulate about the need to register each case of NTM-LD is not understandable. It is not contagious and has no social significance. Of course, it would be helpful to monitor the scale of the problem, but this postulate is not justified in relation to many other more important health problems, where such mandatory registration would be really helpful and necessary.

On what basis did the authors conclude that the cases of NTM-LD were not dependent on population density?

The end of the discussion is too general, too much emphasis is placed on epidemiology and little space is devoted to clinical aspects. It does not indicate which patients require diagnostics towards NTM-LD, what is the effectiveness of diagnostic and therapeutic methods. It is worth expanding these aspects in the discussion.

What is NB disease [319]?

The entire discussion has been thoroughly modified, and in view of the above, the response will be to improve the entire discussion including the removal of some paragraphs that were repetitive or did not affect the harmonious tone of the discussion.

References

References requires profound changes.

Each item is written in a different style. It is necessary to standardize the notation in accordance with the instructions for MDPI authors:

Author 1, A.B.; Author 2, C.D. Title of the article. Abbreviated Journal Name Year, Volume, page range.

The authors use a lot of freedom in writing abbreviations of letters and monograms of names, using dots or not. It is similar with the abbreviations of magazines, sometimes with a dot, sometimes without, and sometimes only at the end of the magazine. It needs to be unified.

Sometimes the volume is in bold, sometimes the year is in bold, and sometimes all numbers are in normal or all numbers are in bold.

Sometimes the name of the authors was entered in full, e.g. poem 444 item 17.

Sometimes there are no spaces between words and often unnecessary numerous spaces, e.g. between monograms of names.

Item 2 was reviewed before the study was undertaken.

Some articles link to Pubmed or Goegel Scholar, while others do not, although they are available.

The number of authors before et al. is very variable. from one, by 3 authors, 6 authors. Sometimes more than 6 authors without introducing the abbreviation et al. It needs to be unified.

In position 43 commas before the surname and not after the monograms. '

Some references use & before the last author, but most are preceded by a comma.

In some articles, all words are capitalized and in others, only the first word in a sentence is capitalized.

The entire references section has been modified.

Reviewer 2 Report

This is an important manuscript, particularly for TB control programs. It highlights a significant issue. A substantial portion of the acid-fast bacteria isolated in the microbiology lab are non-tuberculous mycobacteria (NTM), rather than M. tuberculosis strains. Unfortunately, this information is unavailable in many countries, leading to erroneous treatment of patients with acid-fast bacilli in sputum samples, who are mistakenly prescribed TB drugs.

However, the manuscript requires extensive revision, and an experienced writer should review it. Overall, there is repetition of information within the manuscript. For instance, the abstract mentions three times that there were 149 cases of NTM lung diseases. Furthermore, the abstract contains errors. It states that Mycobacterium kansasii was the most common species (n 24 = 77,52%) in the group, followed by Mycobacterium avium complex (n = 46,31%). However, the last 'n' lacks a number, and the percentages 77.52% and 46.31% exceed 100%.

Moreover, the manuscript could benefit from shortening, and there are figures that could be replaced with text. Additionally, important references from Poland discussing NTM infections have been overlooked. Please include these references and use them in the discussion section.

See:

Nontuberculous mycobacteria strains isolated from patients between 2013 and 2017 in Poland. Our data with respect to the global trends. Adv Respir Med. 2018. doi: 10.5603/ARM.a2018.0047.

Global Environmental Nontuberculous Mycobacteria and Their Contemporaneous Man-Made and Natural Niches. Front Microbiol. 2018 Aug 30;9:2029. doi: 10.3389/fmicb.2018.02029. In this article it has been written that Slovakia, Poland, and the United Kingdom have the highest amount of M. kansasii isolates in Europe (Hoefsloot et al., 2013);

And concerning M. kansasii: Genotyping of human Mycobacterium kansasii isolates from Poland. European Respiratory Journal 2017 50: PA2738; DOI: 10.1183/1393003.congress-2017.PA2738

Please revise the manuscript with a native english speaker

Author Response

We thank the reviewers for their careful review of our manuscript entitled “Trends from the last decade with Nontuberculous Mycobacteria Lung Disease (NTM-LD): clinicians’ perspectives in Regional Center of Pulmonology in Bydgoszcz, Poland”. We have carefully read the comments of the reviewers and have revised our manuscript in light of them.

All co-authors of our manuscript are aware of and agree with the revisions. Detailed point-by-point responses to the reviewers’ comments are provided below.

Thank you once again for considering our manuscript. We look forward to hearing from you regarding our submission.

Point-by-point response to reviewers

1.Faced with suggestions for linguistic improvement. Of course, we completely agree with this comment. The article was consulted by an English specialist, but we concluded that we would ask for an improvement in the language form of the MDPI.

2.The number of key words was increased.

3.As suggested by the reviewers, the entire content of the manuscript was re-examined and a strongly revised version was submitted for further analysis.

4.Due to the rewriting of the manuscript, the numbering of figures and tables has changed

  1. Figure 4 has been replaced by Table 1. We believe that the table significantly improves the way these data are shown
  2. The previous Table 1 (after the change – Table 2) has been reworded. The Table 2 shows underlying conditions, while demographic conditions are described in the results.
  3. Lines numbered in the original will have a different numbering in the revison manuscript.

The introduction has been improved in the hope that it provides background and includes all relevant references now. We have heavily revised the results and the conclusions in the hope that they are now clear and cohesive.

However, the manuscript requires extensive revision, and an experienced writer should review it. Overall, there is repetition of information within the manuscript. For instance, the abstract mentions three times that there were 149 cases of NTM lung diseases. Furthermore, the abstract contains errors. It states that Mycobacterium kansasii was the most common species (n 24 = 77,52%) in the group, followed by Mycobacterium avium complex (n = 46,31%). However, the last 'n' lacks a number, and the percentages 77.52% and 46.31% exceed 100%.

The abstract has been rewritten in accordance with the insights provided in the commentary. The repetitions have been removed. Incorrect numbers and percentages of mycobacteria, resulting from a mistake, have been corrected.

Moreover, the manuscript could benefit from shortening, and there are figures that could be replaced with text.

Figure 4 has been replaced by Table 1. The previous Table 1 (after the change – Table 2) has been reworded. The previous Figure 7 has been replaced with text.

Additionally, important references from Poland discussing NTM infections have been overlooked. Please include these references and use them in the discussion section.

Data on NTM infections from Poland have been added to the introduction. Suggested references have been included in the discussion.

Reviewer 3 Report

Line 64: “Similar data is published in Poland, but it should be noted that accurate records are not available”

If corresponding data are published in Poland, please provide a reference. I suggest adding fact that accurate data are not available because NTM is not a notifiable disease and therefore is no ongoing register.

Line 153: “Once cultured NTM isolate was differentiate from MTBC isolate by rapid immunochromatographic BD MGIT TBc Identification Test (Becton Dickinson) with positive result for MTBC and negative result for NTM.”

This sentence is incomprehensible and needs some amendment.

Figure 6: Please add the percentages in brackets  to the absolute numbers.

Line 206: Please provide the type of “underlying diseases” , e.g. COPD, bronchiectasis.

Line 214: “Precise data about underlying conditions are shown in Table 1.”

The authors offer a mix between “conditions”  and “diseases”. Nicotinism is no underlying disease. Please specify the type of “immundeficiency” in the 7 patients.

The paragraph beginning with line 220 and ending with line 227 lists non-specific symptoms that are not characteristic of NTM and do not contribute to the topic of the article. I recommend that this paragraph and the corresponding figure 7 be deleted in its entirety.

Line 237: “ For the eight years of the research involvement of both lungs was more common than presence of the caves. This proportion has changed in 2021, the next year caves were a more common radiological image of mycobacteriosis (Figure 8).”

What does this statement contribute?

Line 313: “A minimum of 50% of patients with MAC lung disease demonstrate radiographic abnormalities that are characterized by nodules associated with bronchiectasis or NB disease. Nodules and bronchiectasis are observed in the right middle lobe and left lingular segment.”

This statement should have been inserted in the result section, not in the discussion!

The translation into English is - with very few exceptions - generally well understood

Author Response

We thank the reviewers for their careful review of our manuscript entitled “Trends from the last decade with Nontuberculous Mycobacteria Lung Disease (NTM-LD): clinicians’ perspectives in Regional Center of Pulmonology in Bydgoszcz, Poland”. We have carefully read the comments of the reviewers and have revised our manuscript in light of them.

All co-authors of our manuscript are aware of and agree with the revisions. Detailed point-by-point responses to the reviewers’ comments are provided below.

Thank you once again for considering our manuscript. We look forward to hearing from you regarding our submission.

Point-by-point response to reviewers

1.Faced with suggestions for linguistic improvement. Of course, we completely agree with this comment. The article was consulted by an English specialist, but we concluded that we would ask for an improvement in the language form of the MDPI.

2.The number of key words was increased.

3.As suggested by the reviewers, the entire content of the manuscript was re-examined and a strongly revised version was submitted for further analysis.

4.Due to the rewriting of the manuscript, the numbering of figures and tables has changed

  1. Figure 4 has been replaced by Table 1. We believe that the table significantly improves the way these data are shown
  2. The previous Table 1 (after the change – Table 2) has been reworded. The Table 2 shows underlying conditions, while demographic conditions are described in the results.
  3. Lines numbered in the original will have a different numbering in the revison manuscript.

The introduction has been improved in the hope that it provides background and includes all relevant references now. We have heavily revised the results and the conclusions in the hope that they are now clear and cohesive.

Line 64: “Similar data is published in Poland, but it should be noted that accurate records are not available” 

If corresponding data are published in Poland, please provide a reference. I suggest adding fact that accurate data are not available because NTM is not a notifiable disease and therefore is no ongoing register.

The data has been corrected based on our own data and the literature.

Line 153: “Once cultured NTM isolate was differentiate from MTBC isolate by rapid immunochromatographic BD MGIT TBc Identification Test (Becton Dickinson) with positive result for MTBC and negative result for NTM.”

This sentence is incomprehensible and needs some amendment.

The sentence has been thoroughly rewritten. We hope it is understandable now.

Figure 6: Please add the percentages in brackets  to the absolute numbers.

It is Figure 5 now. The percentages have been added.

Line 206: Please provide the type of “underlying diseases” , e.g. COPD, bronchiectasis. 

The authors offer a mix between “conditions”  and “diseases”. Nicotinism is no underlying disease. Please specify the type of “immundeficiency” in the 7 patients.

This line has been improved. We have corrected errors. The previous Table 1 (after the change – Table 2) has been reworded. The Table 2 shows underlying conditions, while demographic conditions are described in the results. The type of immudeficiency in the 7 patients has been specify.

The paragraph beginning with line 220 and ending with line 227 lists non-specific symptoms that are not characteristic of NTM and do not contribute to the topic of the article. I recommend that this paragraph and the corresponding figure 7 be deleted in its entirety.

Following the reviewer's recommendation, paragraph 220 to 227 was eliminated along with figure.

Line 237: “ For the eight years of the research involvement of both lungs was more common than presence of the caves. This proportion has changed in 2021, the next year caves were a more common radiological image of mycobacteriosis (Figure 8).” 

What does this statement contribute? 

The new conclusions have been added.

Line 313: “A minimum of 50% of patients with MAC lung disease demonstrate radiographic abnormalities that are characterized by nodules associated with bronchiectasis or NB disease. Nodules and bronchiectasis are observed in the right middle lobe and left lingular segment.”

This statement should have been inserted in the result section, not in the discussion!

Line 313 has been inserted in the results section.

Reviewer 4 Report

I consider that the article is well structured and represents an epidemiological study of the region, therefore, it should be revised with minor changes.

Lines 20 and 21: Write the full names of the abbreviations, the latter must go between parentheses.

Lines 24 and 25: place the names of the microorganisms in italics.

Lines 29 and 30: Write the full names of the abbreviations, the latter must go between parentheses.

Line 62: Write NTM

Lines 78 and 82: abbreviate Mycobacterium as M.

Line 201: Figure 1 place the names of the Mycobacterium in italics.

Lines 290 and 294: Write the Mycobacterium names in italics.

Author Response

We thank the reviewers for their careful review of our manuscript entitled “Trends from the last decade with Nontuberculous Mycobacteria Lung Disease (NTM-LD): clinicians’ perspectives in Regional Center of Pulmonology in Bydgoszcz, Poland”. We have carefully read the comments of the reviewers and have revised our manuscript in light of them.

All co-authors of our manuscript are aware of and agree with the revisions. Detailed point-by-point responses to the reviewers’ comments are provided below.

Thank you once again for considering our manuscript. We look forward to hearing from you regarding our submission.

Point-by-point response to reviewers

1.Faced with suggestions for linguistic improvement. Of course, we completely agree with this comment. The article was consulted by an English specialist, but we concluded that we would ask for an improvement in the language form of the MDPI.

2.The number of key words was increased.

3.As suggested by the reviewers, the entire content of the manuscript was re-examined and a strongly revised version was submitted for further analysis.

4.Due to the rewriting of the manuscript, the numbering of figures and tables has changed

  1. Figure 4 has been replaced by Table 1. We believe that the table significantly improves the way these data are shown
  2. The previous Table 1 (after the change – Table 2) has been reworded. The Table 2 shows underlying conditions, while demographic conditions are described in the results.
  3. Lines numbered in the original will have a different numbering in the revison manuscript.

Lines 20 and 21: Write the full names of the abbreviations, the latter must go between parentheses.

GenoType NTM-DR, GenoType Mycobacterium AS and GenoType Mycobacterium CM are the names of the assays. There are no abbreviations.

Lines 24 and 25: place the names of the microorganisms in italics.

The names of the microorganisms has been placed in italics.

Lines 29 and 30: Write the full names of the abbreviations, the latter must go between parentheses.

The full names have been written.

Line 62: Write NTM

It has been corrected.

Lines 78 and 82: abbreviate Mycobacterium as M.

It has been corrected.

Line 201: Figure 1 place the names of the Mycobacterium in italics.

The names of the Mycobacterium in the Figure have been placed in italics.

Lines 290 and 294: Write the Mycobacterium names in italics.

The names of the Mycobacterium have been written in italics.

Round 2

Reviewer 1 Report

The paper after the corrections is much better, but still needs corrections.

The summaries, as recommended, have been rewritten and show the content of the article much better.

Keywords completed, but no radiological or clinical aspects yet.

The introduction was shortened and corrected, eliminating numerous errors contained in it.

Minor editorial comments:

Abbreviations were introduced twice: NTM - in keywords and in the introduction, COPD - abstract and introduction, NTM-LD - in keywords and in the introduction.

Some specific information in the introduction still has no source indicated.

I suggest changing the fragment: from 1.2 to 4.38 until 2010 to 4.8 in 2016 [16] [67]

to the original from 1.2 in 2006 to 4.8 in 2016 (per 100,000 patients-year).

The microbiological part of the methodology has been improved, but we still do not know anything about the methodology of assessing radiological changes and clinical data. It is not known what was analyzed and how. Were computed tomography of the lungs assessed in all cases or lung X-rays? The methodology for assessing the clinical changes that were analyzed was not described. This needs supplementing.

It was very beneficial to replace Fig 4 with Table I.

In Table 2, n (%) can be shown more adequately above the relevant columns.

A decrease in BMI was observed in this group?

The authors do not mention anything about treatment and effects. Perhaps this is a topic for another study, but it is worth mentioning.

It is worth presenting the distribution of diagnosed cases each year in the results to be able to talk about the increase in the incidence of NTM-LD. Detailed data on positive cultures per species each year are presented, but we do not know how many cases of disease caused by M kansasii and other mycobacteria are diagnosed each year. It does not allow assessing the incidence to the disease.

It is worth discussing the huge differences between Japan and Poland in the frequency of MAC.

W possibly missing a quote in the sentence A survey from Israel of M. kansasii between 1998 and 2004 showed no association with HIV and high rates of associated lung disease [?] 316

Not only Poland is quoted in the sentence: This predominance is consistent with previous studies from Poland [47,48,49].318

The results presented do not confirm the information in the sentence: the radiological findings of NTM-LD vary depending on the causative organism. 321

The results show an increase in incidence of MAC in the last years and a decrease in M kansasii. This is not shown in the executive summary or conclusions and is a clear epidemiological trend.

The references are still compiled contrary to the instructions for authors. There is a great deal of write freedom. Each item is written differently. It is necessary to unify this record. Position 19 is the same as 48. This needs to be changed.

All in all, the work now requires only minor corrections.

Author Response

We thank for their very careful review of our manuscript entitled “Trends from the last decade with Nontuberculous Mycobacteria Lung Disease (NTM-LD): clinicians’ perspectives in Regional Center of Pulmonology in Bydgoszcz, Poland”. Thank you once again for considering our manuscript. As suggested by one of the reviewers, we agree that after the article is accepted for publication, we will consult the language layer with MDPI specialists.

By answering individual questions, we took up the topic point-by-point response to reviewer

Keywords completed, but no radiological or clinical aspects yet.

Key words were supplemented with radiological and clinical aspects

As suggested by the reviewer, it has been supplemented.

Minor editorial comments:

Abbreviations were introduced twice: NTM - in keywords and in the introduction, COPD - abstract and introduction, NTM-LD - in keywords and in the introduction.

As suggested by the reviewer, corrected.

Some specific information in the introduction still has no source indicated.

As suggested by the reviewer, corrected.

I suggest changing the fragment: from 1.2 to 4.38 until 2010 to 4.8 in 2016 [16] [67]

to the original from 1.2 in 2006 to 4.8 in 2016 (per 100,000 patients-year).

As suggested by the reviewer, corrected.

The microbiological part of the methodology has been improved, but we still do not know anything about the methodology of assessing radiological changes and clinical data. It is not known what was analyzed and how. Were computed tomography of the lungs assessed in all cases or lung X-rays? The methodology for assessing the clinical changes that were analyzed was not described. This needs supplementing.

As suggested, information about methodology of assessing radiological changes and clinical data was added.

It was very beneficial to replace Fig 4 with Table I.

We agree with the reviewer's comment.

In Table 2, n (%) can be shown more adequately above the relevant columns.

Table 2 has been corrected.

A decrease in BMI was observed in this group?

We do not have these data, however, in the preparation of the next publication on this topic, these data will be taken into account and supplemented, as the reviewer's comment clearly presents us with an increase in the value of this work with this subject matter.

The authors do not mention anything about treatment and effects. Perhaps this is a topic for another study, but it is worth mentioning.

Regarding treatment, everything we have collected is already described: the most common set of drugs, radiological and clinical improvement, side effects, treatment time. I think an extended drug analysis would require additional data (203-212).

It is worth presenting the distribution of diagnosed cases each year in the results to be able to talk about the increase in the incidence of NTM-LD. Detailed data on positive cultures per species each year are presented, but we do not know how many cases of disease caused by M kansasii and other mycobacteria are diagnosed each year. It does not allow assessing the incidence to the disease.

According to the sugestion, this section has been modified. The new table has been added in the place of that figure. (Table 2. NTM species among patients with NTM-LD).

It is worth discussing the huge differences between Japan and Poland in the frequency of MAC.

According to the sugestion, it has been added.

W possibly missing a quote in the sentence A survey from Israel of M. kansasii between 1998 and 2004 showed no association with HIV and high rates of associated lung disease [?] 316

As suggested by the reviewer, it was supplemented.

Not only Poland is quoted in the sentence: This predominance is consistent with previous studies from Poland [47,48,49].318

As suggested by the reviewer, corrections and additions have been made.

The results presented do not confirm the information in the sentence: the radiological findings of NTM-LD vary depending on the causative organism. 321

Literature information by other authors added

The results show an increase in incidence of MAC in the last years and a decrease in M kansasii. This is not shown in the executive summary or conclusions and is a clear epidemiological trend.

These details have been added in the Executive Summary and Conclusions

The references are still compiled contrary to the instructions for authors. There is a great deal of write freedom. Each item is written differently. It is necessary to unify this record. Position 19 is the same as 48. This needs to be changed.

References have been corrected and standardized in accordance with the instructions to authors. Items 19 and 48 became one element of the bibliography.

Reviewer 2 Report

I will not oppose against publication. So, approved or " Accept in present form" in the present form. (17th July)

none

Author Response

We thank for their careful review of our manuscript entitled “Trends from the last decade with Nontuberculous Mycobacteria Lung Disease (NTM-LD): clinicians’ perspectives in Regional Center of Pulmonology in Bydgoszcz, Poland”. Thank you once again for considering our manuscript. As suggested by one of the reviewers, we agree that after the article is accepted for publication, we will consult the language layer with MDPI specialists.

Reviewer 3 Report

The authors did a lot of work to address the reviewers´ concerns. The revised version has been improved markedly.

Author Response

(The authors gave the same response as above.)
